# Comparison of antibiotic use and antibiotic resistance between a community hospital and tertiary care hospital for evaluation of the antimicrobial stewardship program in Japan

Mika Morosawa[1,2], Takashi Ueda[3], Kazuhiko Nakajima[3], Tomoko Inoue[4], Masanobu Toyama[4], Hitoshi Ogasiwa[5], Miki Doi[5], Yasuhiro Nozaki[2], Yasushi Murakami[2], Makoto Ishii[1], Yoshio Takesue[3,6]*

1 Department of Respiratory Medicine, Nagoya University Graduate School of Medicine, Nagoya, Japan, 2 Department of Respiratory Medicine, Tokoname City Hospital, Tokoname, Aichi, Japan, 3 Department of Infection Control and Prevention, Hyogo College of Medicine, Nishinomiya, Japan, 4 Department of Pharmacy, Tokoname City Hospital, Tokoname, Aichi, Japan, 5 Department of Clinical Technology, Tokoname City Hospital, Tokoname, Aichi, Japan, 6 Department of Clinical Infectious Diseases, Tokoname City Hospital, Tokoname, Japan

* takesuey@hyo-med.ac.jp

**Data Availability Statement:** All relevant data are within the paper and its Supporting Information files.

## Abstract

Assessment of risk-adjusted antibiotic use (AU) is recommended to evaluate antimicrobial stewardship programs (ASPs). We aimed to compare the amount and diversity of AU and antimicrobial susceptibility of nosocomial isolates between a 266-bed community hospital (CH) and a 963-bed tertiary care hospital (TCH) in Japan. The days of therapy/100 bed days (DOT) was measured for four classes of broad-spectrum antibiotics predominantly used for hospital-onset infections. The diversity of AU was evaluated using the modified antibiotic heterogeneity index (AHI). With 10% relative DOT for fluoroquinolones and 30% for each of the remaining three classes, the modified AHI equals 1. Multidrug resistance (MDR) was defined as resistance to $\geq 3$ anti-*Pseudomonas* antibiotic classes. The DOT was significantly higher in the TCH than in the CH ($10.85 \pm 1.32$ vs. $3.89 \pm 0.93$, p < 0.001). For risk-adjusted AU, the DOT was $6.90 \pm 1.50$ for acute-phase medical wards in the CH, and $8.35 \pm 1.05$ in the TCH excluding the hematology department. In contrast, the DOT of antibiotics for community-acquired infections was higher in the CH than that in the TCH. As quality assessment of AU, higher modified AHI was observed in the TCH than in the CH ($0.832 \pm 0.044$ vs. $0.721 \pm 0.106$, p = 0.003), indicating more diverse use in the TCH. The MDR rate in gram-negative rods was 5.1% in the TCH and 3.4% in the CH (p = 0.453). No significant difference was demonstrated in the MDR rate for *Pseudomonas aeruginosa* and Enterobacteriaceae species between hospitals. Broad-spectrum antibiotics were used differently in the TCH and CH. However, an increased antibiotic burden in the TCH did not cause poor susceptibility, possibly because of diversified AU. Considering the different patient populations, benchmarking AU according to the facility type is promising for inter-hospital comparisons of ASPs.

**Funding:** The author(s) received no specific funding for this work.

**Competing interests:** Y. Takesue received grant support from Shionogi & Co., Ltd. and payment for lectures from Astellas Pharma Inc. and MSD Japan. The other authors have no conflict of interest to declare.

## Introduction

In the era of global antimicrobial resistance (AMR), antimicrobial stewardship (AS) programs have become common in most institutions to reduce antimicrobial selective pressure [1–4]. Concomitant with adequate infection control, AS programs reduce AMR rates at individual institutions [3–6]. AS programs, including prospective audits and feedback as well as formulary restrictions and preauthorization, can effectively decrease antibiotic use in comparison with the pre-intervention phase [7–10]. In general, the relative intra-hospital change in antibiotic use is monitored in the evaluation of AS programs in each hospital. However, chronic excess use of antibiotics can be overlooked until AMR develops only with the monitoring of intra-hospital trend. On the other hand, further decreases in hospital antibiotic use may not be feasible in institutions where comprehensive AS activity has been ongoing for several years. Hence, establishment of uniform standards for the target level of antibiotic use is required. Without proper benchmarks, the implementation of AS programs cannot effectively promote the optimal use of antimicrobials in health care facilities.

Patient populations at tertiary care hospitals (TCHs) have higher degrees of disease severity and are more severely immunocompromised, which together result in more complicated infections in comparison with patients in community hospitals (CHs) [11]. Thus, a greater degree of antibiotic use is imperative in TCHs than in CHs, even with adjustment for bed number. In addition, beds are shared for convalescent phase rehabilitation care in some CHs. Hence, different benchmarks should be determined for TCHs and CHs. Inter-hospital comparisons of risk-adjusted antibiotic use are recommended [12]. To provide a risk-adjusted benchmark of antibiotic use, the Centers for Disease Control and Prevention (CDC) developed the standardized antimicrobial administration ratio (SAAR) [13]. The SAAR is a novel AS metric that compares actual to expected antimicrobial use. However, the SAAR cannot be used worldwide owing to different backgrounds of hospitalized patients in the United States (US) in comparison with patients in other countries, including Japan, as well as different systems of health care and health insurance [14]. Although a Korean SAAR [15] has been developed, such a risk-adjusted benchmark is not available in many countries and this is somewhat complex to calculate for universal use.

In addition to the amount of antibiotic use, several authors have reported the usefulness of antibiotic diversity to prevent AMR [16–19]. Recently, Ueda et al. [20] evaluated balanced use of broad-spectrum antibiotics using the antibiotic heterogeneity indices (AHIs). The authors found a negative correlation between the indices and rate of AMR in *Pseudomonas aeruginosa* and *Klebsiella pneumoniae*. The aim of this study was to compare the amount and diversity of broad-spectrum antibiotic use to assess the performance of AS programs in a CH and a TCH in Japan. The rates of AMR of clinical isolates in these hospitals were also evaluated as outcomes of the respective AS programs. This study represents an early step in developing risk-adjusted benchmarks for antimicrobial use in Japan.

## Methods

This study was conducted at Tokoname City Hospital and Hyogo Medical University Hospital. Tokoname City Hospital is a 266-bed CH (136 beds for acute-phase medical wards) where no intensive care unit (ICU) is present and is one of four designated medical institutions for specified infectious diseases certified by the Japanese government. Hyogo Medical University Hospital is a 963-bed TCH with an ICU and departments that perform organ transplantation procedures including hematopoietic stem cell transplantation, as well as other highly specialized services. The departments in each hospital are listed in S1 Table.

Data were collected from April 2021 to March 2022. This is a retrospective study of medical records or archived samples, and all data were fully anonymized before we accessed them. This study was approved by the institutional review boards at Hyogo Medical University (No. 4194) and Tokoname City Hospital (No 2022–1). Ethics committee waived the requirement for informed consent.

The previous AS team in the CH comprised one pharmacist and two medical technologists in clinical microbiology with 25% full-time equivalent each, with the support of a physician from another hospital who joined AS meetings held once a week. Interventions were carried out for patients with bloodstream infection or meningitis, isolation of multidrug-resistant organisms, and use of antibiotics for longer than 14 days. With the launch of the Department of Infectious Disease, a new multidisciplinary AS program was established in April 2021 at Tokoname City Hospital. The new AS team in this CH comprised one infectious disease physician, one pulmonologist, two pharmacists with 30% full-time equivalent each, three nurses, and two medical technologists in clinical microbiology. Although AS meetings were held twice a week at this CH, an infectious disease physician checked the medical chart every weekday as needed for patients with bacteremia or those with serious infections for whom frequent time-outs were required.

An infection prevention and control department was established in 2006 at Hyogo Medical University Hospital, the TCH in this study. During the study period, the AS program was conducted by one infectious disease physician, a pharmacist dedicated to the AS program, and an infection control nurse, with additional input from a microbiologist. AS rounds were conducted every weekday in this TCH. Prospective audits, feedback, and consultation service with antibiotic timeouts were the main components of the AS programs at both hospitals. Interventions occurred in the following situations: 1) in both hospitals, when input was received from the microbiology laboratory that a patient had bloodstream infection or meningitis or when multidrug-resistant organisms were isolated; 2) when input was received from the pharmaceutical department that broad-spectrum antibiotics with anti-*Pseudomonas* activity were started (TCH: piperacillin/tazobactam and carbapenems; CH: piperacillin/tazobactam, carbapenems, fluoroquinolone, fourth-generation cephems/ceftazidime/aztreonam, and aminoglycosides) or with prolonged use of antimicrobials (TCH: longer than 10 days for carbapenems and pipera-cillin/tazobactam and longer than 14 days for the remaining intravenous antimicrobial agents; CH: longer than 7 days for all intravenous and oral antimicrobial agents). Additionally, the AS program in the CH intervened with the use of anti-methicillin-resistant *Staphylococcus aureus* drugs, antifungal, or antiviral drugs, or drugs needing therapeutic drug monitoring. AS teams in both hospitals supervised antibiotic therapy when clinicians contacted them for advice.

Days of therapy (DOTs)/100 bed days (DOT) were used for the comparison of antibiotic use. We compared DOT values for common intravenous broad-spectrum antibiotics with anti-*Pseudomonas* activity, which were predominantly used for hospital-onset infections including carbapenems, piperacillin/tazobactam, fourth-generation cephalosporins (i.e., cefe-pime and cefozopran)/ceftazidime/aztreonam, and fluoroquinolones. This category of antibi-otics was defined as broad-spectrum antibiotics in this study. The TCH is a high-volume center for allogeneic hematopoietic stem cell transplantation. To decrease the bias in antibiotic use, the DOT at the TCH excluding the division of hematology was also evaluated. Because beds in the CH are shared for acute-phase medical care, convalescent phase rehabilitation care, and community-based integrated care, DOT only for acute-phase medical care wards in the CH was also compared with DOT in the TCH.

In addition to comparisons between the two hospitals, changes in broad-spectrum antibi-otic use were determined in the CH to evaluate the impact of the new AS team. Thus, the pre-(April 2018 to March 2021) and post-intervention periods were compared. The primary study

period compared with the TCH (April 2021 to March 2022) was considered term 1 and the follow-up period after the introduction of the new antibiotic use policy (April 2022 to September 2022) which was proposed based on the previous one year antibiotic use pattern in term 1 was considered term 2. Because the DOT [6] calculation system was not available until March 2021 in the CH, the comparison between pre- and post-establishment of the new AS team was made using the defined daily dose (DDDs)/100 bed days (DDD) [5]. The new assumed average maintenance dose [21] per day for each drug was used for calculation of DDDs.

Using modified categories of intravenous antimicrobial agents reported by the National Healthcare Safety Network [6], we also compared antibiotic use in the TCH and the CH with respect to the following: antibiotics predominantly used for community-acquired infections, including ampicillin/sulbactam, ceftriaxone/cefotaxime, and macrolides; first- and second-generation cephalosporins mainly used for surgical prophylaxis; antibiotics predominantly used for infections with extensively antibiotic-resistant gram-negative organisms; intravenous antibiotics predominantly used for resistant gram-positive infections; intravenous antibiotics with antifungal activity; and antiviral agents.

To assess the heterogeneous use of the four classes of broad-spectrum antibiotics used for hospital-onset infections, we evaluated the relative DOT (% DOT) and the AHIs. The % DOT was calculated as the DOT of one class of antibiotic divided by the DOT of all four classes of broad-spectrum antibiotics. The method of calculating AHIs were demonstrated in S1 Fig. The AHI [18, 22, 23] was calculated as follows:

$$AHI = 1 - \{n/(2 \times [n-1])\} \times \Sigma |a_i - b_i|$$

where n = 4 (the number of antimicrobial classes); $a_i$ = 0.25 for each antibiotic class; and $b_i$ = observed proportion of the DOT for each antibiotic class. The AHI is the index when all antibiotic classes are equally used (% DOT = 25%). When 25% of the relative DOT in each antibiotic class is achieved, the AHI is 1. The modified AHI, which is a new indicator of balanced antibiotic use, was also calculated by defining $a_i$ as 0.1 for fluoroquinolones and 0.3 for each of the remaining three antibiotic class [20]. The modified AHI is the index when each antibiotic class (% DOT = 30%) except for fluoroquinolones (% DOT = 10%) is equally used.

The evaluated organisms were glucose non-fermenting gram-negative rods including *P. aeruginosa* and *Acinetobacter* spp.; Enterobacteriaceae species including *Escherichia coli*, *Klebsiella pneumoniae*, *K. oxytoca*, *K. aerogenes*, *Enterobacter cloacae*, *Enterobacter* spp., *Citrobacter freundii*, *Proteus mirabillis*, *P. vulgaris*, *Serratia marcescens*, and *Morganella morganii*. Clinical isolates from all specimens except stool were used in this study to exclude the influence of surveillance cultures for AMR organisms. The following sampling techniques were recommended in both hospitals. Specimens were collected using strict aseptic technique. Specimens were collected prior to the administration of antimicrobial agents, avoiding contamination with indigenous flora. Swab specimens were unacceptable, with a few exceptions. Specimens were to be delivered at room temperature to the laboratory as soon as possible. If delayed transport was anticipated, urine, stool, and respiratory specimens were refrigerated (not blood specimens). If anaerobic culture was required, anaerobic collection containers were used. Additionally, the following procedures were recommended for blood culture collection. Peripheral blood was collected via venipuncture rather than via indwelling vascular catheter. Blood culture bottle tops were cleaned with antiseptic. Chlorhexidine alcohol was used for skin preparation, allowing the site to dry for at least 30 seconds before venipuncture. Sufficient blood (20–30 mL) was drawn and each bottle was inoculated with 10 mL of the sample; at least two sets were required. Changing needles prior to inoculating blood culture bottles was not recommended because of the risk for needle-stick injury.

Organisms isolated from specimens obtained later than 48 hours after admission were defined as nosocomial isolates. If two or more strains of the same bacterial species were isolated from an individual patient, the most resistant strain was selected for the analysis. Antibiotic susceptibility was determined according to criteria of the European Committee on Antimicrobial Susceptibility Testing (EUCAST) [24] and the Clinical and Laboratory Standards Institute (CLSI) [25] (S2 Table). Resistance according to EUCAST and resistance/intermediate susceptibility according to CLSI for at least two of the following antibiotic classes were analyzed: ciprofloxacin, cefepime, tazobactam/piperacillin, meropenem, and gentamycin/amikacin. MDR was defined as resistant gram-negative rods according to EUCAST (or resistance/intermediate susceptibility according to CLSI) for at least three of the antibiotics listed above.

Isolates of carbapenemase and extended-spectrum β-lactamase (ESBL)-producing Enterobacteriaceae were also compared. A nitrocefin disk (cefinase; BD, Tokyo, Japan) was used to detect β-lactamase production by gram-negative bacteria, according to the manufacturer's instructions. The Cica-Beta Test (Kanto Chemical, Tokyo, Japan) was used to detect ESBL- and metallo-β-lactamase-producing gram-negative bacteria by directly scraping the colony and applying it to the disk. Multiplex nested PCR (BioFire FilmArray panel, Biomerieux Japan, Tokyo) was used to determine the production of carbapenemase.

Categorical variables were compared using chi-squared or Fisher's exact tests. Continuous variables are expressed as mean (standard deviation) and were compared using *t*-tests or Mann–Whitney U tests. Although the DOT or DDD during the period of interest (e.g., one calendar year) was usually monitored, the mean monthly DOT or DDD during a specific time frame was also evaluated for the statistical analysis. The level of statistical significance was set at $p < 0.05$. IBM SPSS ver. 24 (IBM Corp., Armonk, NY, USA) was used for all statistical analyses.

## Results

### Comparison between the TCH and CH regarding antibiotic use and antibiotic resistant rate in gram-negative rods

During the 1-year study period, there were 256,377 patient days in the TCH and 63,033 in the CH. The rates of AS events eliciting input from the microbiology laboratory and from the pharmaceutical department were 59.3% and 29.2%, respectively, in the CH and 17.7% and 27.7%, respectively, in the TCH. The remaining events were consultations from physicians; the rates were 11.5% in the CH and 54.6% in the TCH. Because of a smaller number of hospitalized patients in the CH, wider indications for AS interventions were adopted in the CH than in the TCH; the number of interventions per 100 patient days was 1.25 events in the CH and 0.48 events in the TCH. Timeouts between the initiation and completion of antimicrobial therapy were more frequent in the TCH than in the CH (cumulative number of timeouts per one event: 8.6 and 2.5, respectively), indicating that complicated or refractory infections were treated more often in the TCH.

The mean monthly DOT of the broad-spectrum antibiotics predominantly used for hospital-onset infections was 10.85 ± 1.31 in the TCH, 8.35 ± 1.05 in the TCH excluding the division of hematology, 3.89 ± 0.93 in the CH, and 6.90 ± 1.50 in the acute-phase medical wards of the CH (Table 1). The DOT was significantly higher in the TCH than those in the CH ($p < 0.001$) and acute-phase medical wards in the CH ($p < 0.001$). Even when the division of hematology was excluded in the TCH, the difference was still significant when compared with the CH ($p < 0.001$) and acute-phase medical wards in the CH ($p = 0.012$). The mean monthly DOT of antibiotics used for resistant gram-positive infections and that of anti-fungal agents were also significantly higher in the TCH than those in the CH (Table 1). In contrast, the DOT of antibiotics predominantly used for community-acquired infections was significantly smaller in the

**Table 1. Day of therapy of each category of intravenous antimicrobials in the tertiary care hospital and the community hospital between April 2021 and March 2022.**

| Category of intravenous antimicrobial agents | Day of therapy | | | |
|---|---|---|---|---|
| | Tertiary care hospital | Tertiary care hospital, excluding the division of hematology | Community hospital | Community hospital (acute-phase medical wards) |
| Broad-spectrum antibiotics predominantly used for hospital-onset infections | 10.85 ± 1.31 | 8.35 ± 1.05 | 3.89 ± 0.93[*, §] | 6.90 ± 1.50[*, ‡] |
| Antibiotics predominantly used for community-acquired infections | 5.12 ± 0.62 | 5.39 ± 0.66 | 6.27 ± 1.21[†, ¶] | 11.74 ± 2.50[*, §] |
| Antibiotics used for resistant gram-positive infections | 2.51 ± 0.47 | 1.99 ± 0.33 | 0.38 ± 0.31 [*, §] | 0.63 ± 0.59[*, §] |
| Anti-fungal agents | 1.89 ± 0.46 | 0.52 ± 0.22 | 0.17 ± 0.16 [*, §] | 0.36 ± 0.35[*] |

[*] $<0.001$

[†] p = 0.008 (vs. tertiary care hospital)

[§] $<0.001$

[‡] p = 0.012

[¶] p = 0.038 (vs. tertiary care hospital, excluding division of hematology)

TCH (5.12 ± 0.62) than those in the CH (6.27 ± 1.21, p = 0.008) and acute-phase medical wards in the CH (11.74 ± 2.50, p < 0.001). The DOT throughout the 1-year study period for each category of intravenous antimicrobial agent is presented in S3 Table.

The DOT of each broad-spectrum antibiotic predominantly used for hospital-onset infections and the AHIs for this category of antibiotics are presented in Table 2. A significantly higher modified AHI was shown in the TCH than in the CH (0.832 ± 0.044 vs. 0.721 ± 0.106, p = 0.003), indicating a more heterogeneous (balanced) use of these antibiotics in the TCH (the target of the AHIs was 1). If the division of hematology was excluded in the TCH, a further increase in the

**Table 2. Day of therapy (DOT) in each broad-spectrum antibiotic predominantly used for hospital-onset infections, and antibiotic heterogeneity indices (AHIs) for this category antibiotics in the tertiary care hospital and community hospital.**

| Markers of antibiotic use | Tertiary care hospital | Tertiary care hospital, excluding the division of hematology | Community hospital | Community hospital (acute-phase medical wards) |
|---|---|---|---|---|
| DOT of broad-spectrum antibiotics predominantly used for hospital-onset infections (relative DOT) | | | | |
| Carbapenems | 4.32 ± 0.48 (39.8%) | 2.53 ± 0.28 (30.3%) | 1.13 ± 0.46 (29.1%) | 2.29 ± 1.04 (33.1%) |
| Piperacillin/tazobactam | 3.51 ± 0.55 (32.3%) | 2.93 ± 0.49 (35.1%) | 1.76 ± 0.68 (45.3%) | 3.19 ± 1.12 (46.2%) |
| Fourth-generation cephalosporins, ceftazidime, aztreonam | 2.20 ± 0.43 (20.3%) | 2.11 ± 0.44 (25.3%) | 0.71 ± 0.35 (18.3%) | 1.05 ± 0.71 (15.2%) |
| Fluoroquinolone | 0.82 ± 0.22 (7.5%) | 0.78 ± 0.15 (9.3%) | 0.28 ± 0.20 (7.3%) | 0.38 ± 0.37 (5.5%) |
| AHI (for 25% use in each of the four classes) | 0.704 ± 0.048 | 0.773 ± 0.031 | 0.636 ± 0.113[*, ¶] | 0.569 ± 0.148[§, ¶] |
| Modified AHI (for 30% use in three classes, except for fluoroquinolones [10%]) | 0.832±0.044 | 0.914±0.047 | 0.721±0.106 [†, ¶] | 0.665±0.137 [‡, ¶] |

Mean monthly DOT and AHIs between April 2021 and March 2022 were shown.

[*] p = 0.068

[†] p = 0.003

[§] p = 0.007

[‡] p < 0.001 (vs. tertiary care hospital)

[¶] p < 0.001 (vs. tertiary care hospital, excluding division of hematology)

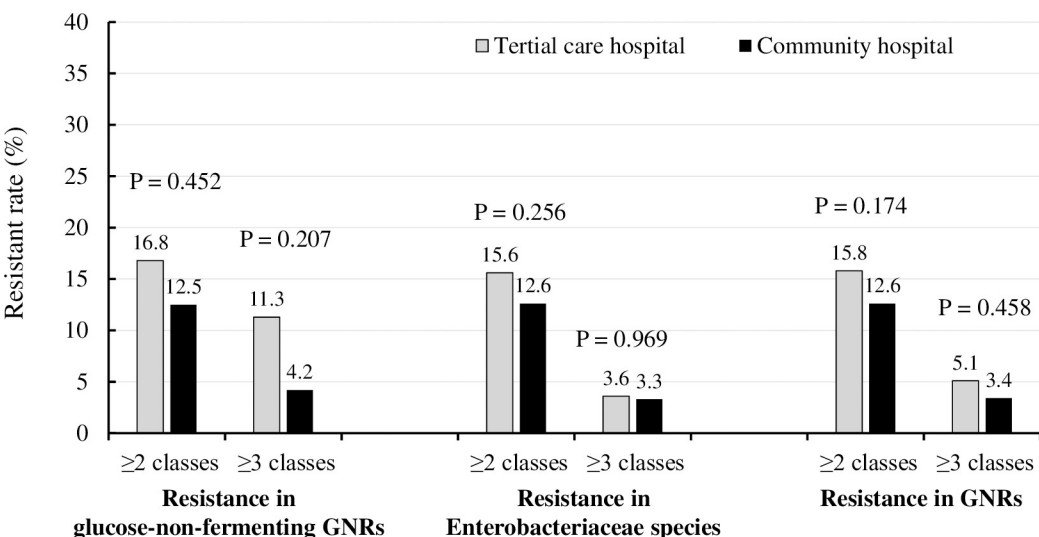

**Fig 1. Resistant rate according to the European Committee on Antimicrobial Susceptibility Testing criteria in nosocomial isolates of Gram-negative rods (GNRs), glucose-non-fermenting GNRs and Enterobacteriaceae species between the tertiary care hospital and community hospital.**

AHIs was observed (AHI, 0.773 ± 0.031; and modified AHI, 0.914 ± 0.047). Excess use of piperacillin/tazobactam (% DOT = 45.3%) caused lower AHIs in the CH. The % DOT of piperacillin/tazobactam was 46.2% when acute-phase medical wards in the CH were evaluated.

In total, 2530 and 559 strains of gram-negative rods were isolated, and 1812 and 262 nosocomial isolates were analyzed in the TCH and CH, respectively. The rate of nosocomial isolates in the TCH was significantly higher than that in the CH (71.6% vs. 46.9%, p < 0.001). In other words, community-acquired infections were treated more frequently in the CH. Resistance rates in gram-negative rods among all nosocomial isolates were similar between the TCH and CH (resistant to ≥2 classes: 15.8% vs. 12.6%, p = 0.174; and resistant to ≥3 classes: 5.1% vs. 3.4%, p = 0.458, respectively). There was no significant difference in the resistance rates among glucose-non-fermenting gram-negative rods (resistant to ≥ 2 classes: p = 0.452; and ≥ 3 classes: p = 0.207) and Enterobacteriaceae species (resistant to ≥ 2 classes: p = 0.256; and ≥ 3 classes: p = 0.969) between these two hospitals (Fig 1).

The results of isolation of resistant strains of each organism according to EUCAST and of non-susceptible strains according to CLSI criteria are shown in Table 3 and S4 Table, respectively. No significant difference in the rate of resistance to ≥ 2 classes (p = 0.413) and to ≥ 3 classes (p = 0.147) was observed for *P. aeruginosa* between the TCH and CH. No *Acinetobacter* spp. strains resistant to ≥ 2 classes were isolated in either hospital. The isolation rates of ESBL-producing *E. coli* were similar (26.7% in the TCH and 25.7% in the CH), and there was no significant difference in the rates of resistance to ≥ 2 classes (p = 0.196) and to ≥ 3 classes (p = 0.948) for *E. coli* between these hospitals. In contrast, the rate of ESBL-producing *K. pneumoniae* tended to be higher in the TCH than the rate in the CH (16.4% vs. 2.9%, p = 0.068). However, the rate of MDR (≥ 3 classes) remained at 7.3% in the TCH.

## Changes in the use of broad-spectrum antibiotics before and after establishment of a new AS team in the CH

In the CH, the DDD decreased significantly, from 3.73 ± 0.87 (p = 0.003), 4.21 ± 0.79 (p < 0.001), and 3.59 ± 0.64 (p = 0.003) in the previous 3 consecutive years, respectively, before

**Table 3. Resistant rate according to the European Committee on Antimicrobial Susceptibility Testing criteria and beta-lactamase-producing rate in each organism in the tertiary care hospital and community hospital.**

| Antibiotic resistant strains | No of resistant strains or β-lactamase-producing strains among nosocomial isolates (%) | | p-value |
|---|---|---|---|
| | Tertiary care hospital (n = 1812) | Community hospital (n = 262) | |
| *Pseudomonas aeruginosa* | n = 296 | n = 43 | |
| ≥2 classes resistance | 61 (20.6%) | 6 (14.0%) | 0.413 |
| ≥3 classes resistance | 41 (13.9%) | 2 (4.7%) | 0.147 |
| *Acinetobacter* spp. | n = 68 | n = 5 | |
| ≥2 classes resistance | 0 (0.0%) | 0 (0%) | – |
| ≥3 classes resistance | 0 (0.0%) | 0 (0%) | – |
| *Escherichia coli* | n = 561 | n = 113 | |
| ≥2 classes resistance | 152 (27.1%) | 24 (21.2%) | 0.196 |
| ≥3 classes resistance | 26 (4.6%) | 6 (5.3%) | 0.948 |
| ESBL producing strains | 144 (26.7%) | 29 (25.7%) | 0.999 |
| carbapenemase producing strains | 0 (0%) | 0 (0%) | – |
| *Klebsiella pneumoniae* | n = 305 | n = 34 | |
| ≥2 classes resistance | 46 (15.1%) | 1 (2.9%) | 0.092 |
| ≥3 classes resistance | 22 (7.3%) | 1 (2.9%) | 0.562 |
| ESBL-producing strains | 50 (16.4%) | 1 (2.9%) | 0.068 |
| carbapenemase-producing strains | 0 (0%) | 0 (0%) | – |
| *Klebsiella oxytoca* | n = 132 | n = 14 | |
| ≥2 classes resistance | 3 (2.3%) | 0 (0%) | 0.674 |
| ≥3 classes resistance | 1 (0.8%) | 0 (0%) | – |
| ESBL-producing strains | 0 (0%) | 0 (0%) | – |
| carbapenemase-producing strains | 0 (0%) | 0 (0%) | – |
| *Klebsiella aerogenes* | n = 68 | n = 7 | |
| ≥2 classes resistance | 2 (2.9%) | 1 (14.2%) | 0.656 |
| ≥3 classes resistance | 1 (1.5%) | 0 (0%) | – |
| ESBL-producing strains | 0 (0%) | 0 (0%) | – |
| carbapenemase-producing strains | 0 (0%) | 0 (0%) | – |
| *Enterobacter* spp.. | n = 190 | n = 21 | |
| ≥2 classes resistance | 19 (10.0%) | 1 (4.8%) | 0.700 |
| ≥3 classes resistance | 1 (0.5%) | 0 (0%) | – |
| ESBL-producing strains | 1 (0.5%) | 0 (0%) | – |
| carbapenemase-producing strains | 0 (0%) | 0 (0%) | – |
| Other Enterobacteriaceae species | n = 192 | n = 25 | |
| ≥2 classes resistance | 4 (2.1%) | 0 (0%) | 0.951 |
| ≥3 classes resistance | 1 (0.5%) | 0 (0%) | – |
| ESBL-producing strains | 2 (1.0%) | 0 (0%) | – |
| carbapenemase-producing strains | 1 (0.5%) | 0 (0%) | – |

ESBL: extended spectrum β-lactamase

the establishment of a new AS team, to 2.62 ± 0.78 in term 1 after establishment of a new AS team (Fig 2). In the evaluation of DDD for acute phase medical wards, there were also significant differences between the pre-establishment period (April 2019–March 2020 [p = 0.003]; and April 2020–March 2021 [p = 0.011]) and term-1 in the post-establishment period.

Frequent use of piperacillin/tazobactam (% DDD: 63.4% between April 2018 and March 2019, 60.6% between April 2019 and March 2020, and 48.2% between April 2020 and March

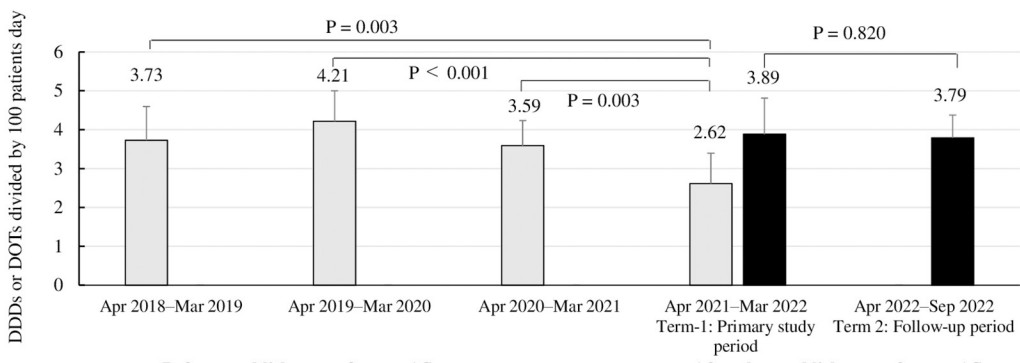

A.  All wards in the community hospital

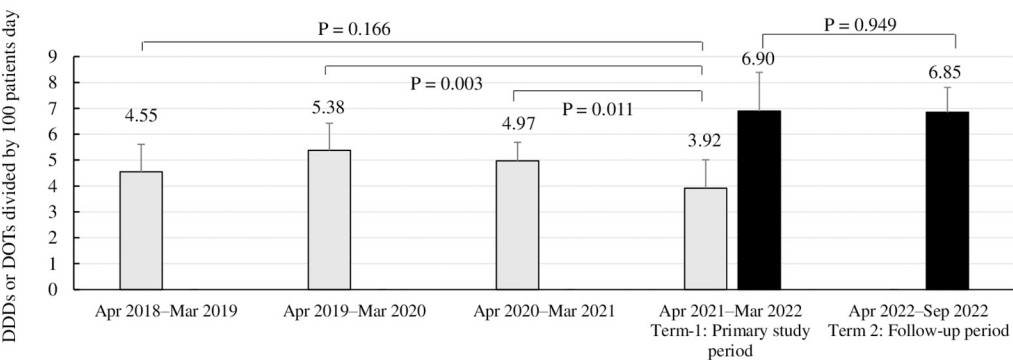

B.  Acute-phase medical wards in the community hospital

**Fig 2. Change of antibiotic use of broad-spectrum antibiotics before and after a new antimicrobial stewardship (AS) team in the community hospital.** DDD: defined daily dose; DOT: day of therapy. Grey bar showed mean monthly DDD and black bar showed mean monthly DOT during the corresponding period.

2021) and infrequent use of fourth-generation cephalosporins/ceftazidime/aztreonam (% DDD: 8.8%, 10.4%, and 10.0%, respectively) were characterized as the antibiotic use pattern before establishment of a new AS team (Table 4). Although the % DDD of fourth-generation cephalosporins/ceftazidime/aztreonam increased to 17.6%, the % DDD of piperacillin/

**Table 4. Defined daily dose (DDD) of each broad-spectrum antibiotic commonly used for hospital-onset infections and the antibiotic heterogeneity indices (AHIs) before and after establishment of a new antimicrobial stewardship (AS) team in the community hospital.**

| Markers of antibiotic use for broad-spectrum antibiotics predominantly used for hospital-onset infections | | Before establishment of a new AS team | | | After establishment of a new AS team |
|---|---|---|---|---|---|
| | | **Apr 2018–Mar 2019** | **Apr 2019–Mar 2020** | **Apr 2020–Mar 2021** | **Apr 2021–Mar 2022 Term 1** |
| DDD (relative DDD) | Piperacillin/tazobactam | 2.37 (63.4%) | 2.63 (60.6%) | 1.73 (48.2%) | 1.17 (44.8%) |
| | Fourth-generation cephalosporins, ceftazidime, aztreonam | 0.33 (8.8%) | 0.45 (10.4%) | 0.36 (10.0%) | 0.46 (17.6%) |
| | Carbapenems | 0.93 (24.9%) | 1.05 (24.2%) | 1.24 (34.5%) | 0.72 (27.6%) |
| | Fluoroquinolones | 0.10 (2.7%) | 0.21 (4.8%) | 0.26 (7.2%) | 0.26 (10.0%) |
| AHIs | AHI | 0.460 ± 0.089* | 0.547 ± 0.133 | 0.558 ± 0.089† | 0.653 ± 0.128 |
| | Modified AHI | 0.558 ± 0.108§ | 0.619 ± 0.121 | 0.667 ± 0.079 | 0.713 ± 0.107 |

* p<0.001

† p = 0.045

§ p = 0.002 (vs. AHIs after establishment of a new AS team)

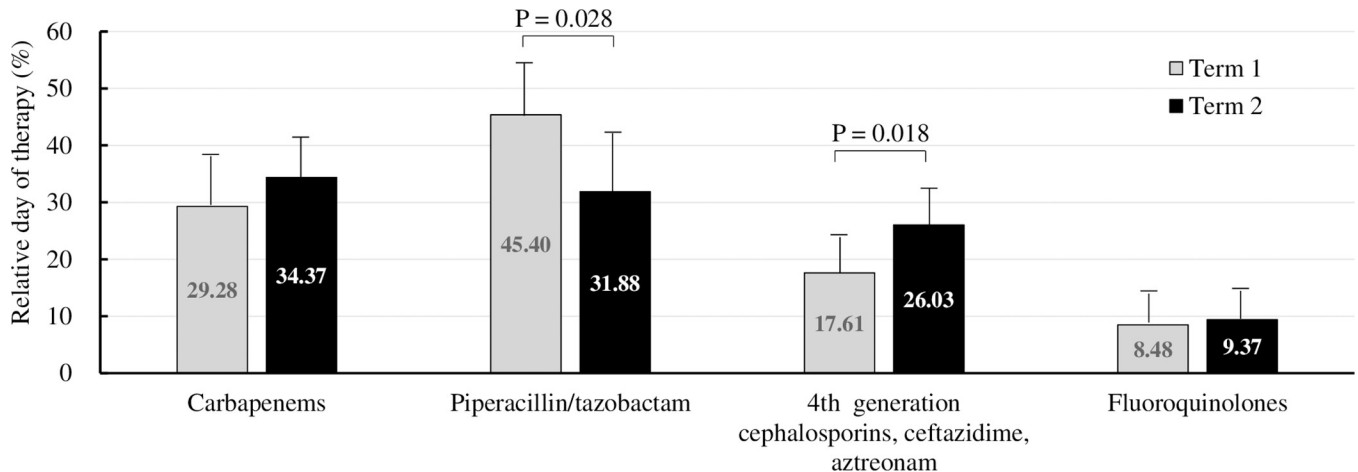

**Fig 3. Relative day of therapy of each broad-spectrum antibiotic commonly used for hospital-onset infections in term 1 and term 2 after establishment of a new antibiotic stewardship team in the community hospital.** Gray bar shows DOT in term 1 and black bar shows DOT in term 2.

tazobactam remained high (44.8%) in term 1 after establishment of the new AS team. A significant improvement in antibiotic heterogeneity was observed only for the AHI (0.653 ± 0.128 vs. 0.558 ± 0.089, p = 0.045) but not for the modified AHI during term 1 in comparison with the preceding 1 year before establishment of a new AS team (Table 4). For the further improvement of antibiotic heterogeneity, the use of piperacillin/tazobactam was discouraged by the AS team and alternative use of fourth-generation cephalosporins was recommended after April 2022 if broad-spectrum antibiotics with anti-*Pseudomonas* activity were required. Interventions by the AS team and the adherence rates in term 2 are presented in S5 Table. The cumulative number of interventions was 344 in 192 events (1.79/event). Discontinuation of antibiotics in patients with prolonged use or non-infectious conditions was a leading reason for intervention (55.7%), followed by alteration of antibiotics other than de-escalation or intravenous to oral stepdown (40.1%). The overall adherence rate was 89.0% (306/344).

Following this policy, the % DOT for piperacillin/tazobactam decreased from 45.40 ± 11.72% in term 1 to 31.88 ± 9.76% in term 2 (p = 0.028), and the % DOT of fourth-generation cephalosporins/ceftazidime/aztreonam increased from 17.61 ± 6.45% to 26.03 ± 6.23% (p = 0.018) (Fig 3), resulting in a significant increase in the AHI (from 0.636 ± 0.113 to 0.746 ± 0.050, p = 0.039) and modified AHI (from 0.721 ± 0.106 to 0.859 ± 0.107, p = 0.019) (Table 5). In the evaluation of acute-phase medical wards in the CH, a significant decrease in the % DOT of piperacillin-tazobactam (from 46.81 ± 15.13% to 31.27 ± 7.63% p = 0.032) and a significant increase in the % DOT of fourth-generation cephalosporins/ceftazidime/aztreonam (from 14.29 ± 8.19% to 25.33 ± 7.22%, p = 0.013) was also achieved in term 2 compared with term 1 (S6 Table). No difference in the total DOT of broad-spectrum antibiotics was observed between the two terms for all wards (term 1: 3.89 ± 0.93 and term 2: 3.79 ± 0.59, p = 0.820) and acute-phase medical wards (term 1: 6.90 ± 1.50; and term 2: 6.85 ± 0.96, p = 0.949) (S6 Table).

**Table 5. Antibiotic heterogeneity indices (AHIs) in term 1 and term 2 after the establishment of a new antibiotic stewardship team in the community hospital.**

| AHIs | Term 1 (Apr 2021–Mar 2022) after establishment of a new AS team | Term 2 (Apr 2022–Sep 2022) after establishment of a new AS team | p-value |
|---|---|---|---|
| AHI | 0.636 ± 0.113 | 0.746 ± 0.050 | 0.039 |
| Modified AHI | 0.721 ± 0.106 | 0.859 ± 0.107 | 0.019 |

## Discussion

We compared AS programs, several markers for antibiotic use, and antibiotic susceptibility between the CH and the TCH in this study. Because of a smaller number of admitted patients in the CH, AS intervention could be more widely indicated in the CH than in the TCH; the number of interventions per 100 patient days was 1.25 in the CH and 0.48 in the TCH, respectively. Similarly, the odds ratio was 5.7 in small- to middle-sized hospitals for conducting interventions within 7 days after the initiation of broad-spectrum antimicrobials [1], compared with large hospitals in Japan. In the present study, even with adjustment for patient days, intravenous antibiotic use was substantially different between the TCH and the CH. The DOT of broad-spectrum antibiotics predominantly used for hospital-onset infections was 3.89 in the CH and 10.85 in the TCH. For risk-adjusted antibiotic use, we also found that the DOT of broad-spectrum antibiotics was 6.90 for acute-phase medical wards in the CH and 8.35 in the TCH excluding the division of hematology.

In an analysis of 576 hospitals in the US (number of beds ≥ 500, 14%), the DOT of anti-*Pseudomonas* drugs was 24.5 [26]. Although oral fluoroquinolones and antibiotics for MDR gram-negative rods were included in the study, a difference in antibiotic use might exist between the US and Japan. The SAAR was developed by the CDC to provide a risk-adjusted benchmark of antibiotic use. Ueda et al. [20] reported the SAAR of broad-spectrum antibiotics predominantly used for hospital-onset infections in a Japanese TCH. Between 2015 and 2022, the SAAR ranged from 1.10 to 1.82 in the ICU, from 0.50 to 1.03 in the department of respiratory medicine, and from 0.70 to 1.20 in the department of colorectal surgery.

In addition to broad-spectrum antibiotics for hospital-onset infections, a significantly higher DOT of antibiotics used for resistant gram-positive infections and antifungal agents was observed in the TCH in comparison with the CH in this study, suggesting frequent empirical and targeted use of these antimicrobial agents for serious infections in the TCH. In contrast, antibiotics predominantly used for community-acquired infections were more frequently used in the CH than in the TCH. These results demonstrate that the DOT in TCHs do not represent the DOT in CHs. In preparation for the development of SAARs in Japan, a nationwide surveillance study is needed to specify the DOT in each antibiotic category according to the following types of hospital for assessment of the AS program in each hospital: small-sized CHs (e.g., < 300 beds) without an ICU, CHs with an ICU, TCHs, and teaching hospitals. The DOT for curative care beds, excluding convalescent rehabilitative care beds, would be useful for standardized risk-adjusted assessment in CHs.

The homogeneous use of a single class of broad-spectrum antibiotics leads to a rapid increase in resistance. Historically, a rotating/cycling empirical antibiotic schedule [27–29] or antimicrobial mixing [26–28] was conducted to achieve the heterogeneous use of broad-spectrum antibiotics in ICUs. Periodic antibiotic monitoring and supervision [23] to facilitate hospital-wide heterogeneous antibiotic use was associated with a decreased rate of resistant gram-negative organisms. The policy of equal use of intravenous broad-spectrum antibiotics (25% in each of the four antibiotic classes) has been reported by several authors [23, 27–29], and the AHI was developed to evaluate whether AS programs achieve the goal of diversified antibiotic use [18, 22, 23].

In a nationwide survey in the US [26], the % DOT of intravenous fluoroquinolones was approximately 23% among the four classes of intravenous broad-spectrum antibiotics. In contrast, the rate of fluoroquinolones among intravenous broad-spectrum antibiotics ranged from 12.4% to 15.4% in an AS program conducted in Japan [23]. Fluoroquinolones were used only in 12.5% of patients with bacteremia caused by difficult-to-treat gram-negative bacilli treated with broad-spectrum antibiotics in a Japanese hospital [30]. Recently, Ueda et al. [20] reported that the % DOT values obtained in a rigorous 7-year AS program were 34.8% for carbapenems,

32.1% for piperacillin/tazobactam, 24.3% for fourth-generation cephalosporins/ceftazidime/ aztreonam, and 8.9% for intravenous fluoroquinolones. From these results, a modified AHI for achieving a % DOT of 10% for fluoroquinolones and 30% for each of the remaining three broad-spectrum antibiotic classes was proposed, and an increased modified AHI value resulted in lower antibiotic resistant rates for *P. aeruginosa* and *K. pneumoniae* [20]. The significant cutoff of the modified AHI to discriminate the risk of antibiotic resistance was 0.899 for *K. pneumoniae* [20].

Contrary to expectations, the widespread use of broad-spectrum antibiotics for hospital-onset infections did not result in poor antibiotic susceptibility of clinical isolates in the TCH. Effective infection control practices [31] and the diversified use of broad-spectrum antibiotics in the AS program may have prevented the emergence and spread of AMR in the TCH. Gandra et al. [32] also found that overall AMR rates in TCHs were not significantly higher than those in small CHs. De-escalation or a rule of conversion to other antibiotics that are effective against the causative organism within 10 days in patients who require prolonged use of carbapenems or piperacillin/tazobactam might facilitate the heterogeneous use of broad-spectrum antibiotics with anti-*Pseudomonas* activity in the TCH. These results indicate that not only quantity assessment of the DOT for broad-spectrum antibiotics but also quality assessment using the AHIs are useful for evaluating AS programs.

After establishing the new AS team in the CH, which included infectious disease physicians dedicated to the AS program, the DOT for broad-spectrum antibiotics decreased compared with the DOT in the pre-establishment term. According to antibiotic use patterns in the initial 1 year (term 1) after establishing the new AS, piperacillin/tazobactam use was discouraged and cefepime was recommended as an alternative in term 2. This policy balanced the use of these broad-spectrum antibiotics within half a year. Antibiotic use was improved in a short period with implementation of a wide range of interventions by the new AS team; this pattern may characterize AS programs in CHs with smaller patient numbers than TCHs if the target goal is clearly specified and rigorous AS activity is conducted to achieve the target.

There are several limitations to this study. First, this was a retrospective study comparing only one institution of each type. Second, this study was conducted during a period with a high prevalence of infections owing to severe acute respiratory syndrome coronavirus 2, which might have affected antibiotic use in the study hospitals [33], especially the CH, which is one of four designated medical institutions for specified infectious diseases in Japan. Finally, few infectious disease physicians who are specialists certified by the Japanese Association of Infectious Diseases are dedicated to AS programs in CHs. Hence, the DOT and the AHIs in this study may not represent small- or mid-sized CHs.

In conclusion, significant differences in antibiotic use in each antibiotic category were found between the CH and TCH in this study. However, increased broad-spectrum antibiotic use did not cause higher resistance rates in the TCH than those in the CH, possibly because of antibiotic diversity. To evaluate the effectiveness of AS programs, applying benchmarks of antibiotic use according to facility type and hospital size are a promising approach.

## Supporting information

**S1 Table. The comparison of patients days in each department between Hyogo medical university hospital and Tokoname city hospital.**
(PPTX)

**S2 Table. MIC breakpoints based on EUCAST and CLSI for Pseudomonas aeruginosa, Acinetobacter spp., and Enterobacteriaceae species.**
(PPTX)

**S3 Table. Day of therapy (DOT) throughout the study period (April 2021 to March 2022) of each category of intravenous antimicrobial agent in the tertiary care hospital and the community hospital.**
(PPTX)

**S4 Table. Rate of non-susceptibility among nosocomial isolates according to the Clinical & Laboratory Standards Institute criteria between the tertiary hospital and the community hospital.**
(PPTX)

**S5 Table. Reasons for intervention by antimicrobial stewardship team and adherence rate by physicians between April 2022 and September 2022.**
(PPTX)

**S6 Table. Days of therapy (DOT) and relative DOT and antibiotic heterogeneity indices (AHIs) for broad-spectrum antibiotics commonly used for hospital-onset infections on the acute-phase medical wards in the community hospital.**
(PPTX)

**S1 Fig. Method of calculating antibiotic heterogeneity indices (AHIs).**
(PPTX)

## Author Contributions

**Conceptualization:** Makoto Ishii.

**Data curation:** Yoshio Takesue.

**Formal analysis:** Takashi Ueda, Yoshio Takesue.

**Funding acquisition:** Yoshio Takesue.

**Investigation:** Mika Morosawa, Takashi Ueda, Kazuhiko Nakajima, Tomoko Inoue, Masanobu Toyama, Hitoshi Ogasiwa, Miki Doi, Yasuhiro Nozaki, Yasushi Murakami.

**Methodology:** Yoshio Takesue.

**Project administration:** Takashi Ueda, Yoshio Takesue.

**Resources:** Yoshio Takesue.

**Software:** Yoshio Takesue.

**Supervision:** Makoto Ishii.

**Validation:** Yoshio Takesue.

**Visualization:** Yoshio Takesue.

**Writing – original draft:** Mika Morosawa, Takashi Ueda, Yoshio Takesue.

**Writing – review & editing:** Mika Morosawa, Takashi Ueda, Yoshio Takesue.

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
