## [Decision Letter · Decision Letter 0]

3 Mar 2023

PONE-D-22-30723A comparison of antibiotic use and antibiotic resistance between a community hospital and a tertiary care hospital for the evaluation of the antimicrobial stewardship program in JapanPLOS ONE

Dear Dr. Takesue,

Thank you for submitting your manuscript to PLOS ONE. After careful consideration, we feel that it has merit but does not fully meet PLOS ONE’s publication criteria as it currently stands. Therefore, we invite you to submit a revised version of the manuscript that addresses the points raised during the review process.

We look forward to receiving your revised manuscript.

Kind regards,

Kwame Kumi Asare, Ph.D

Academic Editor

PLOS ONE

Journal Requirements:

Y. Takesue received grant support from Shionogi & Co., Ltd. and payment for lectures from Astellas Pharma Inc. and MSD Japan. The other authors have no conflict of interest to declare. 

Reviewers' comments:

Reviewer's Responses to Questions

**Comments to the Author**

1. Is the manuscript technically sound, and do the data support the conclusions?

Reviewer #1: Yes

Reviewer #2: Yes

2. Has the statistical analysis been performed appropriately and rigorously? 

Reviewer #1: Yes

Reviewer #2: Yes

3. Have the authors made all data underlying the findings in their manuscript fully available?

Reviewer #1: Yes

Reviewer #2: Yes

4. Is the manuscript presented in an intelligible fashion and written in standard English?

Reviewer #1: Yes

Reviewer #2: No

5. Review Comments to the Author

Reviewer #1: About this study

The assessment of risk-adjusted antibiotic use (AU) is recommended to evaluate the antimicrobial stewardship program. In this respect, in this study, authors aimed to compare the amount and diversity of AU and antimicrobial susceptibility of nosocomial isolates between a community hospital and a tertiary care hospital as a reality in Japan.

Hard points

This manuscript could be of interest for the Journal's public.

What to improve / add

- In the Introduction part, please add more info about the necessity of standards uniformization and why so imperious necessary to develop risk-adjusted benchmarks for antimicrobial use.

- The comparative statistics for the facilities analyzed would be more suggestive as "T-bars" graphic with P values attached.

Reviewer #2: A comparison of antibiotic use and antibiotic resistance between a community hospital and a tertiary care hospital for the evaluation of the antimicrobial stewardship program in Japan

• In the abstract, the THC hospital bed size needs to be mentioned, the sampling techniques are missing, and the total number of resistant isolates needs to be mentioned.

• In the methodology part, you need to mention the sampling techniques.

• The authors mentioned the pre- and post-intervention periods that were compared. So, before discussing the new AS programs, it is necessary to discuss what kind of program they had in the CH and THC.

• For the resistant isolates, the EUCAST values are missing, which need to be considered for optimal therapy.

• The tables should be consistent across all manuscripts.

• The mentioned table (S2 Table) needs more clarification to represent the comparison. I rephrased the sentence to make it more understandable.

• The overall manuscript needs to be checked for grammatical mistakes.

Good Luck!

6. PLOS authors have the option to publish the peer review history of their article (what does this mean?). If published, this will include your full peer review and any attached files.

Reviewer #1: No

Reviewer #2: No

---

## [Author Response · Author response to Decision Letter 0]

29 Mar 2023

We appreciate the time and effort that the editor and the reviewers dedicated to providing feedback on our manuscript and are grateful for the insightful comments on and valuable improvements to our paper. We have incorporated all suggestions made by the reviewers. Please see following Table for a point-by-point response to the reviewers’ comments and concerns.

Main change point

1. Additional statistically analysis was performed using mean monthly antibiotic use markers (DOT, DDD, AHIs) 

2. We also evaluated antibiotic use in the acute-phase medical wards (excluding rehabilitation beds) for the community hospital in comparison with the tertiary care hospital

The data in the acute-phase medical wards of the CH was included in Table 1, Table 2, Figure 2 and S6 Table. 

Line 267-270: The mean monthly DOT of the broad-spectrum antibiotics predominantly used for hospital-onset infections was 10.85 ± 1.31 in the TCH, 8.35 ± 1.05 in the TCH excluding the division of hematology, 3.89 ± 0.93 in the CH, and 6.90 ± 1.50 in the acute-phase medical wards of the CH (Table 1). 

Line 276-279: the DOT of antibiotics predominantly used for community-acquired infections was significantly smaller in the TCH (5.12 ± 0.62) than those in the CH (6.27 ± 1.21, p = 0.008) and acute-phase medical wards in the CH (11.74 ± 2.50, p < 0.001).

Line 382-386: In the evaluation of acute-phase medical wards in the CH, a significant decrease in the % DOT of piperacillin-tazobactam (from 46.81 ± 15.13% to 31.27 ± 7.63% p = 0.032) and a significant increase in the % DOT of fourth-generation cephalosporins/ceftazidime/aztreonam (from 14.29 ± 8.19% to 25.33 ± 7.22%, p = 0.013) was also achieved in term 2 compared with term 1 (S6 Table). 

Reviewer 1

1. In the Introduction part, please add more info about the necessity of standards uniformization and why so imperious necessary to develop risk-adjusted benchmarks for antimicrobial use.

→Line 76-91 (Introduction)

In general, the relative intra-hospital change in antibiotic use is monitored in the evaluation of AS programs in each hospital. However, chronic excess use of antibiotics can be overlooked until AMR develops only with the monitoring of intra-hospital trend. On the other hand, further decreases in hospital antibiotic use may not be feasible in institutions where comprehensive AS activity has been ongoing for several years. Hence, establishment of uniform standards for the target level of antibiotic use is required. Without proper benchmarks, the implementation of AS programs cannot effectively promote the optimal use of antimicrobials in health care facilities.

Patient populations at tertiary care hospitals (TCHs) have higher degrees of disease severity and are more severely immunocompromised, which together result in more complicated infections in comparison with patients in community hospitals (CHs) [11]. Thus, a greater degree of antibiotic use is imperative in TCHs than in CHs, even with adjustment for bed number. In addition, beds are shared for convalescent phase rehabilitation care in some CHs. Hence, different benchmarks should be determined for TCHs and CHs.

2. The comparative statistics for the facilities analyzed would be more suggestive as "T-bars" graphic with P values attached.

→In initial submitted version. antibiotic use throughout the corresponding period (1 year) was evaluated. In revised version, mean monthly antibiotic markers was evaluated for the statistically analysis. 

Line 247-249: Although the DOT or DDD during the period of interest (e.g., one calendar year) was usually monitored, the mean monthly DOT or DDD during a specific time frame was also evaluated for the statistical analysis.

"T-bars" graphic with P values were attached in Figure 2, and P value was added in Figure 1 and Figure 3.

Reviewer 2

1. In the abstract, the THC hospital bed size needs to be mentioned, the sampling techniques are missing, and the total number of resistant isolates needs to be mentioned.

→Line 50 (abstract): a 963-bed tertiary care hospital (TCH) 

Line 62-63: The MDR rate in gram-negative rods was 5.1% in the TCH and 3.4% in the CH (p = 0.453).

Because of limited word count, the sampling techniques could not be included in the abstract. Sampling technique was mentioned in the method.

2. The authors mentioned the pre- and post-intervention periods that were compared. So, before discussing the new AS programs, it is necessary to discuss what kind of program they had in the CH and THC.

→Because we found the poor antibiotic diverse use was the problem in the CH based on the comparison with the TCH, we started a new policy for antibiotic use (decrease the use of piperacillin/tazobactam, and increase the use of 4th generation cephalosporins) in the CH. So. pre- and post-intervention periods were compared for the CH to demonstrate the early response of AS program. This may characterize AS program in CHs with smaller patient numbers than TCHs if the target goal is clearly specified and rigorous AS activity is conducted to achieve the goal.

Line 126-130: The previous AS team in the CH comprised one pharmacist and two medical technologists in clinical microbiology with 25% full-time equivalent each, with the support of a physician from another hospital who joined AS meetings held once a week. Interventions were carried out for patients with bloodstream infection or meningitis, isolation of multidrug-resistant organisms, and use of antibiotics for longer than 14 days.

3. For the resistant isolates, the EUCAST values are missing, which need to be considered for optimal therapy.

→

Line 229-230: Antibiotic susceptibility was determined according to criteria of the European Committee on Antimicrobial Susceptibility Testing (EUCAST) [24]

Rate of resistant strains was evaluated according to EUCAST criteria in Figure 1 and Table 3. Non-susceptibility rate by CLSI was moved to S4 table. Definition of MIC breakpoints based on EUCAST and CLSI was demonstrated in S2 table. 

Accordingly, antibiotic susceptibility was demonstrated based on EUCAST in the result part.

4. The tables should be consistent across all manuscripts.

→The tables were amended, and the same style was used across the manuscript.

5. The mentioned table (S2 Table) needs more clarification to represent the comparison. I rephrased the sentence to make it more understandable.

→According to reviewer’s comment, we divided the S2 table to Table 4 for DDD comparison between pre-intervention phase and term 1 in post-intervention, and Figure 3 for DOT comparison between term 1 and term 2 in post-intervention.

6. The overall manuscript needs to be checked for grammatical mistakes.

→The revised manuscript was checked by a native speaker with medical knowledge. Analisa Avila, MPH, ELS, of Edanz (https://jp.edanz.com/ac) edited a draft of revised manuscript.

---

## [Decision Letter · Decision Letter 1]

10 Apr 2023

Comparison of antibiotic use and antibiotic resistance between a community hospital and tertiary care hospital for evaluation of the antimicrobial stewardship program in Japan

PONE-D-22-30723R1

Dear Dr. Takesue,

We’re pleased to inform you that your manuscript has been judged scientifically suitable for publication and will be formally accepted for publication once it meets all outstanding technical requirements.

Kind regards,

Kwame Kumi Asare, Ph.D

Academic Editor

PLOS ONE

Additional Editor Comments (optional):

Reviewers' comments:

Reviewer's Responses to Questions

**Comments to the Author**

1. If the authors have adequately addressed your comments raised in a previous round of review and you feel that this manuscript is now acceptable for publication, you may indicate that here to bypass the “Comments to the Author” section, enter your conflict of interest statement in the “Confidential to Editor” section, and submit your "Accept" recommendation.

Reviewer #1: All comments have been addressed

Reviewer #2: All comments have been addressed

2. Is the manuscript technically sound, and do the data support the conclusions?

Reviewer #1: Yes

Reviewer #2: Partly

3. Has the statistical analysis been performed appropriately and rigorously? 

Reviewer #1: Yes

Reviewer #2: I Don't Know

4. Have the authors made all data underlying the findings in their manuscript fully available?

Reviewer #1: Yes

Reviewer #2: No

5. Is the manuscript presented in an intelligible fashion and written in standard English?

Reviewer #1: Yes

Reviewer #2: Yes

6. Review Comments to the Author

Reviewer #1: Now this manuscript is acceptable for publishing in the PlosOne Journal. Al my concerns were amended.

Reviewer #2: Authors addressed my comments accordingly and in proofreading is needed specially to check grammar twice.

7. PLOS authors have the option to publish the peer review history of their article (what does this mean?). If published, this will include your full peer review and any attached files.

Reviewer #1: No

Reviewer #2: **Yes: **Faiz Ullah Khan

---

## [Editor Report · Acceptance letter]

14 Apr 2023

PONE-D-22-30723R1 

Comparison of antibiotic use and antibiotic resistance between a community hospital and tertiary care hospital for evaluation of the antimicrobial stewardship program in Japan 

Dear Dr. Takesue:

I'm pleased to inform you that your manuscript has been deemed suitable for publication in PLOS ONE. Congratulations! Your manuscript is now with our production department. 

Kind regards, 

on behalf of

Dr. Kwame Kumi Asare 

Academic Editor

PLOS ONE